# Reframing Kiruna's Relocation—Spatial Production or a Sustainable Transformation?

**Aslı Tepecik Diş** [1,*] **and Elahe Karimnia** [2]

1   School of Architecture and the Built Environment, KTH Royal Institute of Technology, 10044 Stockholm, Sweden
2   Theatrum Mundi, London EC1R 0AA, UK; elahe@theatrum-mundi.org
*   Correspondence: atdis@kth.se; Tel.: +46-737681373

**Abstract:** Due to the expansion of nearby mining operations, the city of Kiruna, an arctic city in Sweden, has been undergoing a massive urban transformation, led by the mining company, Luossavaara-Kiirunavaara Aktiebolag (LKAB), which is the largest iron ore producer in the EU. This paper explores this relocation in a three-sphere transformation framework that has sustainability as the outcome (practical sphere), and analyses it as a socio-spatial transformation process, including political decisions as its driving forces (political sphere), to examine how this outcome and decisions represent individual and collective values (personal sphere). The analysis of three spheres is used as a tool to understand how and why Kiruna's urban transformation is deemed to be sustainable, as it claims, and which it is being globally acknowledged for. Methods include analysis of Kiruna's new master plan, media representations, and interviews with key actors of the project, who include municipal planners; the mining company's planning developers; consultants, as the designers of 'Kiruna 4-ever' and the new city center; as well as the city's residents. The analysis is a critique of the approaches that fit this project into either the critique of market-led spatial production, or as an example of best practice, based on its participatory processes. Results indicate that although Kiruna's relocation is claimed to be a transformation of collective values, practical and technical transformations were dominant, which represents only partial responses in the framework. Therefore, a multi-voice narrative challenges the sustainability of Kiruna's transformation.

**Keywords:** urban transformation; planning practice; urban design process; socio-spatial process; sustainability; urban planning; sustainable transformation

## 1. Introduction

### 1.1. Kiruna's Relocation Background

Urban transformation in arctic cities is largely driven by resource-intensive (extractive) industries. Noticeably, less attention has been paid to what this means for urban sustainability in the circumpolar north [1,2]. This creates a challenging position concerning the view of the overall sustainable development of these cities, as these industries' operations are associated with a range of sustainability issues, including negative local environmental and land-use impacts, emissions of greenhouse gases and socioeconomic tensions. These effects become important from a local perspective, since the concentration of resource-intensive sectors in specific regions plays a very large role for some local and regional economies [3]. The sustainability of urban environments in this region is uncertain, when cities are built on the basis of exploiting a local natural resource, as they face the risk of not being able to reinvent themselves once the resource is depleted ([1], p. 10). It is thus important to address the relationship between urban planning and the sustainable development of resource-dependent communities in this region and understand the processes through which this relationship is being designed and implemented.

Kiruna, a city located in the north of Sweden, 145 km north of the Arctic Circle, is an example of mine-driven urbanization. It belongs to Sweden's northernmost county,

Norrbotten. The area was originally inhabited by the indigenous Sami people and the land was used for reindeer herding. The construction of the railway in 1903, between Kiruna and the Norwegian town of Narvik, started the emergence of a mining community in Kiruna. Hjalmar Lundbohm, a geologist named as Kiruna's founder, was appointed as the first managing director of the state-owned mining company, Luossavaara Kiirunavaara Aktiebolag (LKAB). From the beginning, Kiruna was used as a test site, by aiming high; it was planned to be a model town [4]. The best architects, artists and urban planners were hired to build the city, with a futuristic vision of an arctic city. Since then, Kiruna has grown to be home to over 23,000 inhabitants, many of whom are employees of the mine, the largest underground iron ore mine in the world.

While the idea of a city on the move is not entirely novel, it is likely to become increasingly common, due to the effects of our changing climate and its differentiated impacts around the globe. Given the lessons learned and their global significance, it is important to identify and discuss how Kiruna's spatial production and the process of its transformation might be different, since it claims that it is being sustainable, particularly in the context of communities that are going to be displaced.

Kiruna's urban transformation is often discussed in the context of deliberative planning processes, while the relocation as such is referred to as an opportunity to design a completely new, model city, with a unique way of working with sustainable urban planning, as well as sustainable urban transformation ([5], p. 12, [6], [7], pp. 75–97, [8–10]). Initially, our main aim for the study was to document the planning process for this complex case and understand what made it unique and exemplary in terms of its sustainable planning practices. We also wanted to understand how such a megaproject, with diverse stakeholders and interests, has managed the process without any major conflicts, despite many uncertainties, and since it was framed as a model city for sustainable urban transformation. We aimed at presenting lessons learned that could be of use for other similar cases around the world. However, as we proceeded with the analysis of our material, we noticed that there were different voices that challenged this single narrative that was constructed around the sustainability of the new city. Therefore, our research question resulted in asking: How can we understand Kiruna's urban transformation and its sustainability, not only as an outcome, but as a process?

### 1.1.1. Kiruna's Relocation: Planning and Timeline

Sweden has a highly decentralized policy in which regional and local authorities are granted considerable autonomy, with the national government providing the framework and structure for local government activities. In principle, the local municipalities have full authority over planning: they have the task of determining the use of land and water within a legal framework that is supervised by the national government [11]. The process of developing a comprehensive plan (översiktsplan) entails collaboration and dialogue with residents and it is usually presented to the public. Planning processes for spatial development require public consultation by law and the citizens have formal opportunities to comment on proposed plans through community consultation processes. The legal aspects of consultations are dealt with in different legal codes; including the Environmental Code, the Planning and Building Act, and the Road Law. All point out that the responsible authority is required to consult the County Administrative Board (Länsstyrelsen, in Swedish) and individuals presumed to be particularly affected by the project, in the early stages of planning ([12], p. 159). However, how these processes should be carried out, practically speaking, is not specified, leaving plenty of room for how communication should be organized, and often regulated by the particular directives for each official agency ([12], p. 157).

Swedish municipalities play a 'proactive' role in planning practice, exercising active political leadership, formulating alternative strategies, and promoting creative dialogue ([12], pp. 11–21). The engagement of local authorities in the planning process connects economic and growth-oriented policies to governance. The planning regulations should

be flexible enough to be influenced by requirements and individual decisions ([12], pp. 157–165). Given the wide range of local authority responsibilities in Sweden, a critical understanding of pragmatic planning processes is crucial to investigate individual decisions and influences [13]. On the other hand, "*the local government has limited influence on land use defined by property rights, specifically in the mountain municipalities where the State controls virtually all land (mountains and forest land) in the western areas*" ([14], p. 43). The situation with the relocation of Kiruna's city center is also subject to this condition, which the politician whom we interviewed in the Kiruna municipality emphasized. Since the national government owns most of the land and there are very few private owners, when an industrial company needs to be moved to a new location, the municipality has to buy the land from the national government and is also constrained by the national interests (Interview with Informant 4). Kiruna's urbanization is also embedded in 11 national interests, some of which include grazing lands for Sami reindeer, national highways, the railway and airport infrastructure ([15], p.68).

The planning process in Kiruna is described as being unique in Swedish urban planning, which often takes a long time to adopt. However, in Kiruna's case, due to time pressure, the existing city already had to begin to be phased out in 2016. Because the new homes and business premises needed to be ready for occupancy, the process has had to proceed quickly—but without sacrificing the quality as specified in the plan. It is this reality that formed the conditions for the work on the new development plan (Interview with Informant 3).

Eventually, in Spring 2004, LKAB submitted a request for amendment of Kiruna's comprehensive plan, due to the expansion of mining operations beneath Kiruna's settled areas and towards central parts of the city. The expansion was expected to generate severe risk of subsidence and damage to the infrastructure. In December 2004, the municipal council voted to produce a new Detailed Development Plan for central Kiruna. In January 2007, the new Detailed Development Plan was adopted, including distinct statements concerning the democratic character of the process ([16], p. 436). The opportunity for all the inhabitants to influence the relocation process has been always emphasized, promising an open and accessible process [16] (read more in Appendix A).

In September 2011, the municipal council adopted a decision on a new location for the urban center. The immediate decision about an extraordinary urban relocation process was made: the city was to be moved 3 km east to guarantee the residents' safety and safeguard the continuation of the mining. In September 2012, the municipality reached out to the Swedish Association of Architects. The council approved an international competition, between ten architecture firms, for a design of a 20-year masterplan to be fulfilled by 2033. The aim of the competition was to create a sustainable, distinctive and pleasant urban environment, a city center linking together the surrounding housing and industrial areas with the rest of the city and constituting the natural hub of the new Kiruna. This would be an opportunity for creating something completely new, emanating from Kiruna's unique history, to accommodate future needs and the desire for good quality of living in an arctic climate. [4]

The winning proposal, by White Architects in collaboration with Ghilardi + Hellsten Architects, was entitled "Kiruna 4-ever"; it offered a sustainable vision for the long-term expansion of the city eastwards (the time plan and the density evolution during 100 years is illustrated in Figure 1). The Detailed Development Plan, completed in 2014, included the new urban structures; it expanded urban densities and offered new public transport systems and new urban form. Overall, it had little relation to the original Kiruna [17], but emphasized its democratic values, such as integrity, equity, inclusion, justice, and ethical and normative sides of transformations [5]. It aimed for broadening Kiruna's diverse cultures and population by creating an attractive and global city within the arctic landscape, while allowing the further development of the mine. The new plan placed a strong focus on civil dialogue. The actual process of moving started in 2017 and was expected to finish in 2033 [18]. As the timeframe of the relocation shows, it took the city

almost 10 years, from the time LKAB warned local politicians to either move parts of the city centre or risk losing Kiruna's largest employer and, accordingly, the future of the whole city, to make the decision.

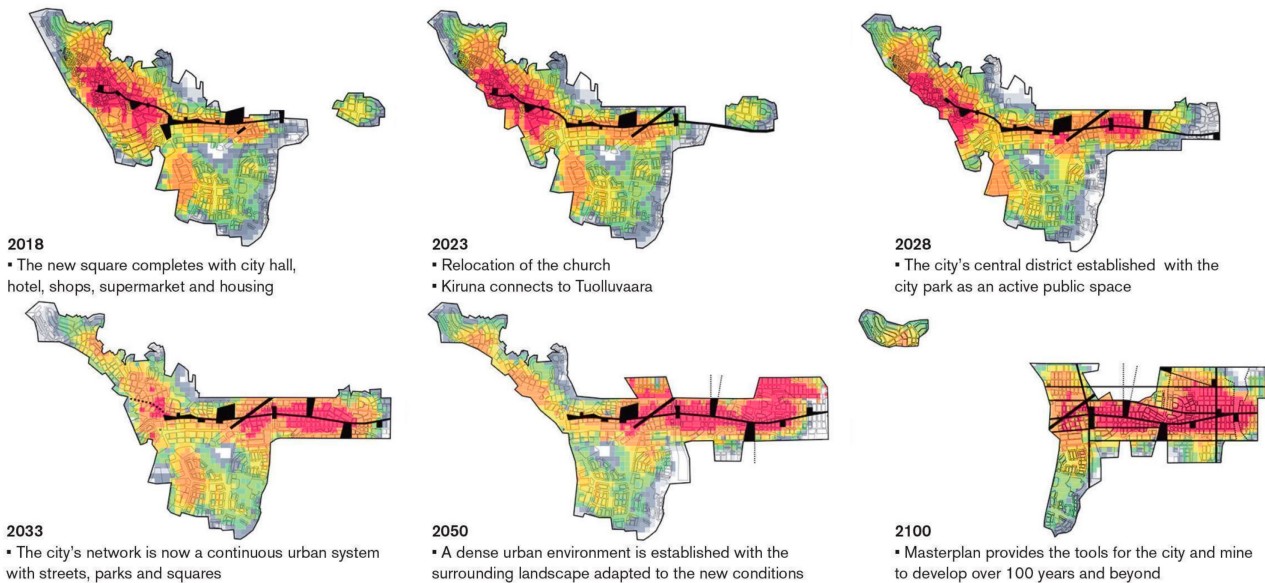

**Figure 1.** The time plan of Kiruna's relocation and density evolution. The diagram is part of the competition entry from 2012. Source. Spacescape, Ghilardi+Hellsten and White Architects.

From the beginning, Kiruna's relocation was framed by the city as a future model for moving other cities ([16], p. 439, [19]). It is acknowledged globally as unprecedented, and as the first real-world example of the relocation of a town of its size [20]. The municipality claims that there has been a good planning process, namely one that has expressed the local authority's sensitivity towards authenticity and the relocation of built heritage [21]. The planning process also emphasized citizen's dialogue and the promise of providing tools for people to take part in the development process and contribute to the changing of their city.

However, Kiruna does not serve as a model in terms of financing, since, according to Swedish law, the mine is paying for the relocation. This draws attention to the mining and its large-scale economic impact as the direct driving force for Kiruna's urban transformation. In order to understand the position of LKAB in Kiruna's transformation, the next section discusses the mining from a national perspective and considers the collaboration between the municipality and the mining company as facilitators of the relocation.

### 1.1.2. The Political Power of LKAB

Mineral extraction and metals processing have been key components of the Swedish economy for most of the country's history ([3], p. 8), both in terms of job creation and their share in total business-sector added value. Overall, rising global iron prices are the main driving force of the expansion of the mining operations and its major subsidence, both of which result in a significant part of the city's move to keep both the city safe and continue the iron ore mining.

While the political structure grants the municipality full planning power over the comprehensive plan, Kiruna faced the unique stakeholder of the mining company, LKAB, who holds the de facto power over the town: "*In a realistic perspective, the mining company has the power over the local authority, even if the local authority has the official power through the planning monopoly*" [15]. Therefore, Kiruna municipality's role in planning practice, and in exercising political leadership, formulating strategies, and promoting creative dialogue ([4], p. 21) requires further critical analysis to understand the relocation as a transformation with socio-spatial production processes.

### 1.2. Kiruna's Transformation Process

### 1.2.1. The Uncertainties

Complexity in planning refers to uncertainties and unpredictable outcomes, such as when different natural, technical and social conditions are integrated with actions and reactions from various actors and stakeholders; and when a great number and variety of elements and time dimensions interact in society as a whole and in planning in particular [15]. Kiruna's transformation was met by the mining company with a strongly constructed narrative and communicated to the residents as the only practical response, with no space to discuss the necessity of it or the uncertainty of its process [16]. The uncertainty of Kiruna's relocation as an outcome is related to the demand for and global price of iron ore, which might change and thus affect future mining activities and, accordingly, the relocation [15].

The strong relocation narrative and related statements, however, could not justify the uncertainties of the planning process. For example, in 2006 concerns were raised by both the municipality about the built heritage and authenticity; these were followed up by LKAB's (see [14] review of the planning documents, their purpose and decisions, between 1984 and 2014 ([21], p. 113–114)). The early decisions, in 2006, show a clear intention to maintain characteristic features of the town by moving listed buildings and in designing new ones. But the planning for the new Town Hall and the city centre reveals a shift in the decisions regarding built heritage and authenticity. The historic buildings are spread out in newly planned areas and Kiruna's new urban form has little relation to the original Kiruna [21].

Similarly, Sandberg and Rönnblom [22] highlight how the relocation as a hegemonic discourse articulated what should be expressed. They raise concerns about the inevitability of people's emotions about the uncertainty of their future and their reactions and responses to the prevailing uncertain and overwhelming situation. These emotions were studied in relation to the points in the urban transformation process, such as "*planning the new city center, and knowing very little about compensation levels and practicalities regarding the redemption process of individuals' houses*" ([22], p. 53). They suggest, considering the unclear process and timeframe of Kiruna's relocation, that this process should be understood through power relations in planning and the design of space and time.

This should be done with questions such as "*When particular types of development should take place, whose time frames should be prioritized (and who is marginalized) in the development process, and how practices should be governed and regulated.*" ([23], p. 2670). The timeframe of transformation projects, according to Raco, et al. [23], affects the perception of their legitimacy. For example, managing community expectations and aspirations can define the form and character of a project. Urban transformation therefore should be perceived and assessed as the result of large-scale political decisions that shape action timeframes, particularly for socio-spatial production processes [24].

### 1.2.2. Reframing the Narrative

There is a tendency in urban planning and design to exclude the how and why questions of the process of spatial production and to focus explicitly on outcome, while urban transformation through a critical lens is argued through spatial political economy and the power relations involved in the process of spatial production [23,24]. Through this perspective, the spatial production of cities, both as outcome and process, is discussed in relation to circuits of capital and as the outcome of ongoing change in systems of capital production ([24], pp. 61–67). Analyzing urban transformation requires understanding of its long-term impact. It is recommended to place its understanding within the larger context of its common use ([24], p. 212). Therefore, we attempt to provide a wider framework in order to understand transformation through different scales and perspectives. Here, we present a case study of planning practice of Kiruna that is interpreted with the help of the three spheres of transformation framework proposed by O'Brien and Sygna [25]. This framework was originally designed by Sharma [26] and adapted by

O'Brien and Sygna [25] as an investigative device that captures transformation in three different spheres. Transformation is referred to as changes in physical [27] or qualitative form, structure [28], or meaning-making [29], and also as a psycho-social process involving the unleashing of human potential to commit, care, and effect change for a better life [30] (p. 4). Stirling [31] writes about emancipating transformations, though not directly in an urban transformation context, but reflecting on transformation processes from a perspective of ideological strategies. He argues that positive social transformation occurs in a reciprocal caring relationship, rather than in the hierarchies of control.

Drawing on Sharma's [26] work, O'Brien and Sygna [25] conceptualize transformation in relation to sustainability and as a process that takes place across three embedded and interacting spheres (Figure 2). Hence, the notion of three spheres of transformation can be interpreted as a tool for understanding deliberate transformations to sustainability. Referring to "practical" (changes in form), "political" (changes in structure) and "personal" (changes in meaning making) spheres of transformation can help capture both the breadth and depth of changes needed to realize a particular goal or an outcome ([25], p.4). The practical sphere refers to behavioral changes, social and technological innovations, and institutional and managerial reforms. The political sphere includes the systems, structures, rules, regulations and structure of information flows that create the conditions for transformations in the practical sphere. The personal sphere includes individual and collective beliefs, values and worldviews that shape the ways that the systems and structures (i.e., the political sphere) are viewed, and influence what types of solutions (e.g., the practical sphere) are considered "possible" and influence the way we see and relate to systems as individuals, groups and cultures ([25], p. 6).

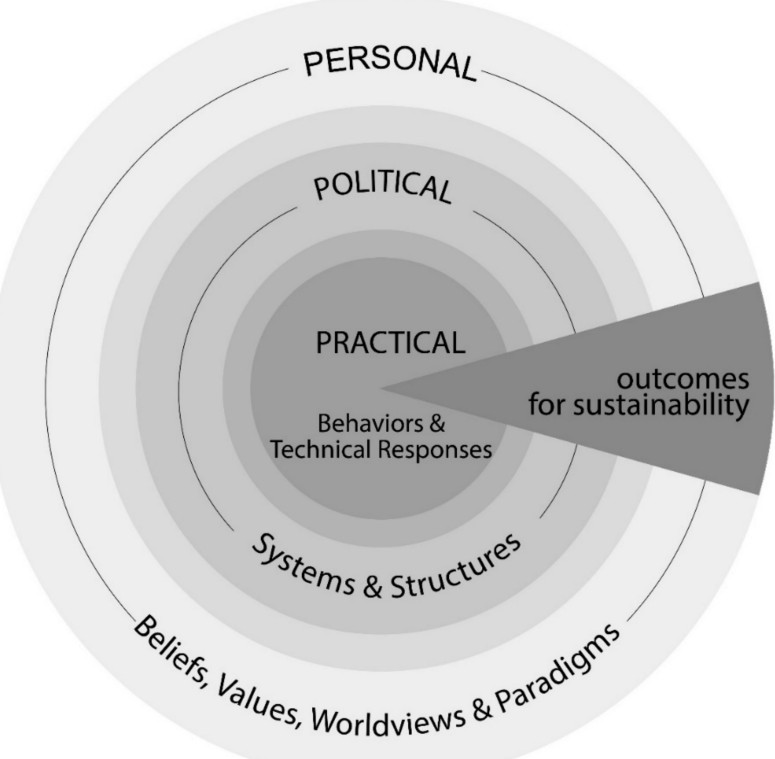

**Figure 2.** The three spheres of transformation (after Sharma [26]) by O'Brien and Sygna [25], illustrated by the authors.

These three spheres of transformation are implicitly or explicitly alluded to in each of the conversations on transformation, but with little attention to their interactions and interrelations ([25], p. 5). It is possible to see the depth of transformations, as well as the multiple entry points for sustainability outcomes, if all these three spheres can be

viewed together and in relation to each other. These spheres are embedded within one another. The ordering of the spheres is stated as being important; the practical sphere being at the core, where the targets or goals are located; the political sphere representing the enabling/disabling conditions; and the personal sphere capturing individual and collective "views" of systems and solutions. Fragmented ways of dealing with the three spheres of transformation, illustrated in Figure 2, would result in partial responses, especially when the responses do not engage in all of the spheres. For reaching outcomes for sustainability, there is a need to work with the practical, political and personal spheres at the same time.

Kiruna's relocation, following O'Brien and Sygna's [25] conceptualization, is explored here as the relation between the three spheres: how the goals and targets of the socio-spatial transformation (the political sphere) have enabled or constrained citizen's lives (the practical sphere), while addressing their individual and collective values (Personal sphere). We use this framework to understand Kiruna's transformations in a wider sense, but also the starting points, such as the "democratic values" claimed in the New Development Plan, to see how the process engages from values to the results.

Through this framework, our study attempts to contribute to the knowledge on sustainable transformation in an urban context and provide a new lens to assess the sustainability of urban transformation projects in different spheres as processes.

## 2. Materials and Methods

Kiruna's urban transformation is regarded as "*one of the largest urban transformations of our time*" [6,7,32]. An urban transformation project of this magnitude has been of interest to us as researchers in the field of urban planning since it has provided the opportunity to study "*a contemporary phenomenon within its real-world context*" ([33], p. 33). As Campbell [34] notes, "*It is difficult for urbanists to detach phenomena from context because it is this context itself that is the subject of study*" ([34], p. 2). Kiruna's relocation, as our case study, takes place within a certain institutional framework and requires local knowledge to realize the needs for these projects. As a process, it involves multiple stakeholders with diverse interests within a specific context that has implications for the qualitative methodological approach. Campbell [35] argues that "*The nature of planning gives the case study approach many advantages over other methodologies because case studies can help understand complex urban processes with unclear boundaries, inputs and outputs.*" ([35], p. 1).

Campbell [34] highlights the importance of the choice of cases, which determines the type of generalization that follows (p. 9). He represents a broader distinction between cases as "typical" and "exceptional." He identifies exceptional cases as 'more effective for challenging existing analytical assumptions and pushing theory forward' and puts them in four different categories: apart from 'critical cases' (a single case disproves an assumption by proving that something is indeed possible), there are 'prescient cases' (phenomena ahead of the time), 'exaggerated cases' (extreme phenomena that happen in few places) and deviant cases (abnormal conditions) ([34], p. 9). Following Campbell's categories, Kiruna is an extreme case, because it is home to the largest iron ore mine in the world, run by the state-owned company that is responsible by law for financing the relocation of the city, whose two-thirds of its population are dependent upon the mine for employment. Previous research claims that Kiruna's relocation is an extreme and complex case [15,36] considering its long-term planning vision, yet under pressure of the time for development. Kiruna's planning process involves many actors with complex interactions. There are different natural, technical and social conditions integrated with the actions and reactions from various sectors when a great number and variety of elements and time dimensions interact in society as a whole and in planning in particular ([36], p. 63). Complexity, as Nilsson discusses, is not only related to multiplicity of actors, but is also about uncertainties and unpredictable outcomes.

We based our analysis on a single case study [33,37] and employed an in-depth examination of the process in order to understand the project's complex nature. To be able to conduct a case study, it is vital to consider what the questions of the study are,

what data is relevant, what type of data needs to be collected and how, and also how the results are to be analyzed [33]. The urban transformation of Kiruna has often been portrayed as a sustainable process leading to a more sustainable city ([15], p. 33), [5,6]). Following our theoretical framework, the questions concerning the overall sustainability of the transformation process guided our data collection procedure. The questions go beyond this particularly strong and positive narrative—the new city being destined to be a model city for sustainable development—constructed and communicated to the public.

During our case study, we used different sources of evidence [33] to understand the process and stakeholders' positions towards Kiruna's urban transformation. The transformation was explored further following the theoretical framework, which analyses sustainability of outcome as a connection between responses in political, practical and personal spheres of transformation process. This framework has given us the opportunity to analyze different phases of transformation for sustainability through the lenses of key stakeholders involved in the process. Our methodology provides a larger and multi-scalar understanding of the narrative of the relocation and its timeframe, including the voices of the following: LKAB, proposing, financing and implementing the relocation of Kiruna; the local spatial planners at Kiruna Municipality, who have strong ties with and dependency on LKAB for the relocation process while facilitating the public engagement and leading the process; the White architects, as consultants (in cooperation with the municipality), designing the masterplan of the New Kiruna; as well as Kiruna residents, who are directly affected by this process, whether they move to the new city or not.

The data upon which this paper is based was collected between 2015 and 2018. A qualitative approach was applied, with a wide range of materials involving an explorative and evaluative analysis. Our empirical data consists of both primary and secondary sources. Initially, the data collection started with direct and participant observation during a site visit to Kiruna in 2015 and followed by two additional site visits in 2016 and 2018. On the first site visit, we joined a large group of researchers to visit a Sami village and engaged in direct observation of a different setting than a city atmosphere. A Sami village representative hosted the group and we had overt participant observation, and listened to the Sami representative's reflections concerning the impacts of mining development and how it affected the Sami community. During our first site visit, we also made initial contacts with Kiruna Municipality and listened to a planner's presentation of undergoing urban transformation in Kiruna. We also engaged in direct observation of the city's major social places, in the city centre. On the second and third visits, we carried out some of the interviews with residents and planners. As a final step, we collected the secondary data, through text analysis of planning documents, namely, the new development plan of Kiruna; the winning proposal for the new Kiruna vision of Kiruna 4-ever; information provided by the websites of LKAB, Kiruna Kommun and White Architects as well as other local and international media representations of the project; and the newspaper articles for the period of the research, to highlight how the New Kiruna is presented to its existing residents. This step was used to verify certain aspects for the interviews and to develop a better understanding of the current planning situation in Sweden, as well as of the transformation process in relation to the dominant narrative present. As part of this last step, we have also followed a Facebook group titled "Vi som sörjer att Kiruna rivs" (trans., we who mourn the destruction of Kiruna), established by some of the residents of Kiruna. The members of the group share their personal feelings, mainly their grief concerning the city's transformation and the loss of their past and their feelings of uncertainty for the future. The group had 1400 members and used the visual and textual elements of Kiruna's urban transformation in their posts.

The main data was collected through 16 "key informants" ([33], pp. 183–184), whose roles were critical in our case study and who gave us access to other interviewees. We carried out a combination of semi-structured interviews of thirteen people and a survey whereby three people responded to questions via email. Accordingly, nine informants were the key professionals from three major bodies: Kiruna Municipality; White Architects,

who were selected by Kiruna Municipality and responsible for assisting with the new development plan; and the mining company, LKAB. Two of the informants were academics: one of them was involved in the practice of the project as a practitioner. Six of our informants were ordinary residents. We had a combination of face-to-face and telephone interviews, all conducted in English, while only one of the face-to-face interviews was in Swedish. Table A1 shows more detailed information about the interviews and the informants (please see Appendix B).

Interviews were conducted between 2016 and 2018, a period after the vision and design for the new city were accepted and when its implementation process had started to take place. The selection of the 6 residents was carried out randomly; however, we paid attention to having respondents who would represent different parts of Kiruna society. The selection of the informants was purposive sampling of nine professionals who represent the strategic actors in the planning and implementation of the transformation process. We also conducted an interview with an academic who lives in Luleå, but works with projects at Kiruna Sustainability Center. The duration of the interviews was approximately 1 h and up to 75 min. The face-to-face interviews were conducted in Kiruna, in participants' workplaces; only one of them took place in Kiruna's tourist office. The interviews were tape-recorded with the permission of the respondents and transcribed later. The interviews were carried out by two researchers (the authors of this paper) separately, yet analysis and interpretation of data were carried out jointly. The informants involved are anonymized, considering the sensitive nature of the topic.

The interviews were undertaken using a semi-structured approach, in order to allow the participants to reflect upon the transformation process as freely as possible, which helped us determine the coding of the themes that emerged. We used an interview guide to cover the main questions informed by the overarching focus of the research, including the planning process and visions for the future of Kiruna, and how these were being addressed through transformation. The interview questions were slightly modified depending on the interviewee, as when talking to professionals the questions were focused on visions and priorities concerning the task of city transformation and, when talking to ordinary residents, more on personal reflections, expectations and reactions concerning transformation. This process helped by allowing space for the participants to provide their side of the story of the transformation and gave us the opportunity to understand what is highlighted in their reflections in relation to the overall narrative that was widely presented in the reports, as well as media. Accordingly, we did a thematic analysis to identify patterns of themes in the interview data, which is reflected in the result section, with a multi-voice approach.

### 3. Results: A Multi-Voiced Narrative of Kiruna's Transformation

*3.1. The Relocation's Facilitationsand Negotiations*

Kiruna's urban transformation process is defined as a development game, characterized by having many issues at stake, and involving stakeholders with conflicting interests (interview with Informant 3). Mutual interdependence existed between stakeholders: they include LKAB, the County Administrative Board, politicians, real estate owners, the Swedish Transport Administration, investors, developers, residents and businesses.

The division of tasks in facilitating the move and negotiations with residents reveals partial responses to transformation by the political planning system. For example, the municipality of Kiruna works on the new development plan (practical sphere of transformation), while LKAB is responsible for the communication with the residents affected by the move and to discuss the timing for the relocation, as well as the compensation measures based upon the valuation of the buildings (personal sphere of transformation) [38] (more details in Appendix C). The timeline of the transformation was steered by LKAB as they followed the timeline of the mining operations. The municipality had to follow and adapt to this timeline, which created pressure on the process as a whole [39].

Another partial response from the municipality and LKAB is related to conservation of the built heritage and authenticity of Kiruna, an issue that was discussed prior to and

during the relocation. The municipality, LKAB and White Architects placed emphasis on moving buildings that were considered to be Kiruna's architectural signature and cultural heritage. The plan included moving some of the cultural identity and cultural landmarks to the new city centre, and mixing some of the old buildings or incorporating parts of them into the new structures (such as the relocation of the clock tower of the old city hall next to the new city hall). Moving these buildings, as illustrated in Figure 3, occupied significant space in the media in terms of the technology used during this process. However, the result shows they were only partial actions and responses [21] that considered the technical and financial limitations of moving all listed buildings. The shift in the decisions made between LKAB and the municipality shows the lack of relation between the personal and practical spheres of transformation.

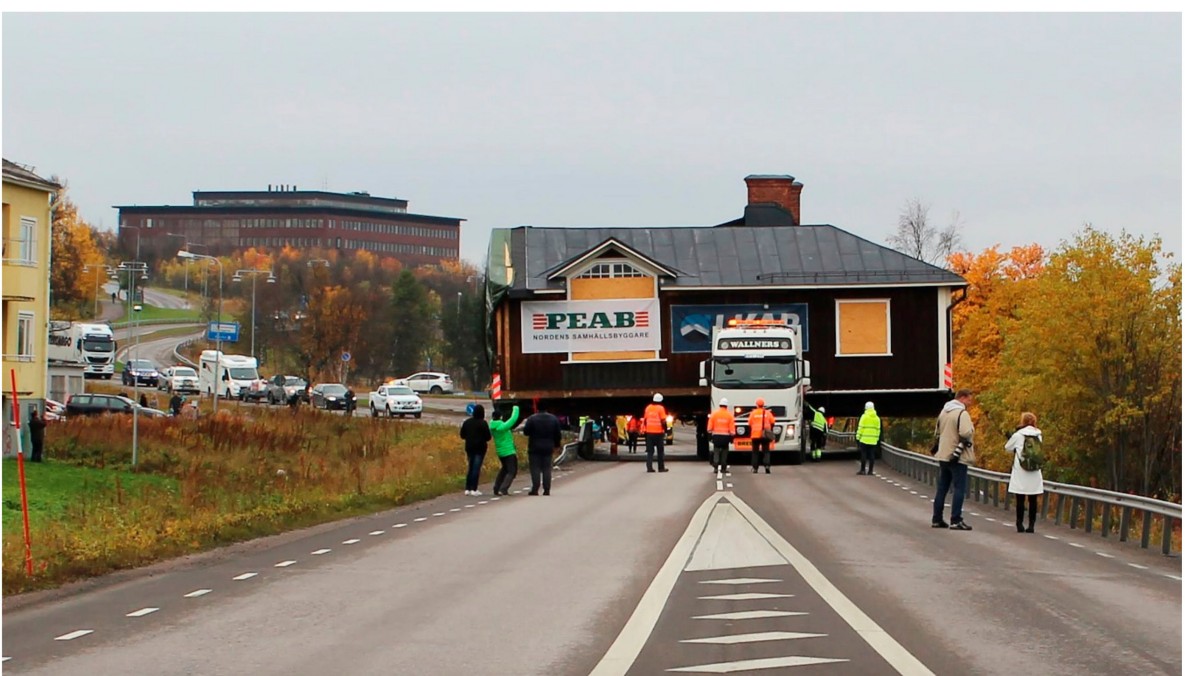

**Figure 3.** Relocating one of the old town's heritage buildings. Credit: Kiruna Municipality.

In the planning process, residents were defined as key actors and were asked about their expectations for the new city centre, in terms of what design elements should be included in the new plan. Altogether, a main square; a shopping street, as a shared space and offering meeting-places; and a dense structure with mixed functions were some of the elements highlighted by the residents and planners, respectively. The planning, claimed as a deliberative process, was reduced to a spatial production process, by means of inquiring about residents' wishes and expectations for the new city design. Hence, the claimed democratic transformation planning process started with the practical sphere, as technical responses with spatial production. This shift highlights the pragmatic approaches that have been gradually adopted by the municipality along with the relocation process.

In terms of deliberative process, the municipality had to test different forms of public consultation processes, as the public was hesitant to talk openly during big public information meetings. The Municipal official mentioned that these big meetings tended to be dominated by older men who think their opinions represent the Kiruna residents as a whole (interview with Informant 3). The municipality decided to focus on the quality of the communication, rather than the quantity of people gathered in the big public meetings, since they did not receive as much information from as many people as they expected. The politician from Kiruna Municipality reflected upon the difficulty of receiving feedback from residents:

> *"It was not easy to understand what they thought and what they wanted. We also did surveys by sending out questions. So we got feedback from people in Kiruna in different ways but it was difficult to get a response. We tried to reach out to the associations instead. When you get to the people in small groups, it is much easier to get them to talk"* (interview with Informant 4)

Accordingly, Kiruna Municipality decided to have focus groups including trade unions, culture organizations, women's meeting clubs, tourist organizations, culture organizations and senior citizens organizations. Planners stated the importance of understanding the wishes of young women and young families for the new city design, as having them stay would be important for the future of Kiruna.

It was interesting to observe that the transformation project is strongly highlighted as a socio-spatial production process by means of engaging residents as a whole in the process of developing the new city centre. This part of the strategy was linked to social sustainability of transformation (Kiruna 4-ever plan). However, the same level of enthusiasm for engaging the residents did not seem to be visible in the decision-making process regarding whether the relocation should take place or not. The decision-making process in Kiruna left no space for creating an organisational context in which the decision was made, but, instead, the decision was made in an "exceptional circumstance," due to the fact that there is only one option, one reality.

Hence, there has always been one possible narrative concerning the decision about the move: the mining activities have to continue and the city has to move. It is impeccably clear that the mine is the foundation of Kiruna in many ways, quite literally as well as economically and structurally. Any threat to this foundation would seem to lead to dire consequences for the city, its residents and LKAB. The narrative of "residents showed a great understanding for the move" has been the norm:

> *"People are used to seeing that we somehow adapt ourselves according to the development of the mine. People in Kiruna are dependent on the mine and they know that if we (the city) continue with the mining, we need to move. Maybe they are not happy but they accept it"* (interview with Informant 4)

### 3.2. The Politics of Time

Although the discussions on where and how to move the buildings took a very long time, the decision about the relocation was made quickly, because of the unquestionable acceptance of the understanding that "without the mine the town would not exist." Therefore, suggested plans and proposals did not receive much public opposition, apart from a few members of a political party. Whether or not there would be a relocation was not a question, yet, as the planners at LKAB showed, its process was their main concern:

> *"For us, of course, our aim with moving a city is not actually to move the city, it is to get the ore. We can't move the ore but we can only move the city. And could we choose? [...] And that is important that we do this in the best way possible; considering human rights in the end by making sure that we are transparent."*

The mining is considered as a sustainable practice, and its impact based on national environmental assessments:

> *"We have really high quality, our product has less emissions than any of our competitors. We have a country that controls emissions and the labor rights and the system as a whole. Sweden is a good country to mine in"* (interview with Informant 8)

The mining company is officially in charge of the relocation's timeline and developing priorities in the process of the move. Thus, for practical reasons, once the plan was out, in 2014, there was not much time left before the existing city plan was to be phased out in 2016. Due to time pressure, planners had to focus more on the implementation of the the new city center's new development plan than on discussion. The planner from LKAB explained, as follows:

*"Once the decision was taken about the new location for the city center, time table for the new city center was so tight so it was almost like [a] don't plan and just do it kind of attitude had to take place"* (interview with Informant 7)

The actual process of moving started on the 24 May 2017, when the new homes and businesses were promised to be ready for occupancy. The mining company [the department of social transformation at LKAB] deals with every case individually, contacting every household to ask about their wishes about the relocation, and to negotiate whether they want a new apartment or money to buy somewhere else, so the households are in a queue, waiting to be contacted, and, as we heard from Kiruna municipality's head of the relocation, they keep a 1-year buffer between the move and when the actual impact of the mine on their housing appears. These timelines, from the practical and technical aspects, are justified, but lack sufficient consideration of collective and individual demands and values, as we mention in the following section.

The timeline of the relocation follows the mining development plan and the intention of accelerating the process. Unlike urban development projects that need to attract private developers to invest in the project and seek profit directly, as discussed by Raco, et al. [23], the mining company as the developer of the project has a long-term engagement in the process, with a longer-term goal and investments, which is mining. Also, the capital investment of the mining company in this project (Kiruna's relocation) has other sources and reasons. The faster the decision is made and the move is carried out, the higher stability for their economic productivity and the employers, who are the future inhabitants of the project. So, for the mining company, Kiruna's relocation as a project is not the goal, but rather a precondition; an important step in achieving their bigger project, which is the mining extension [40].

Kiruna's mining activities will not be forever, so planning for 'Kiruna 4-ever' is a political strategy, which promises Kiruna's citizens that they will have a timeless right to their city [41]. The politics of time, however, highlight that the decisions are made based on market demand (political economy), rather than on a democratic process.

*3.3. Kiruna 4-Ever*

After winning the proposal, White Architects opened an office in Kiruna and a team from Stockholm moved there to occupy it. In collaboration with LKAB and the municipality's planning developers, they developed the new master plan, following seven disciplines, referring to the 'attractiveness' of the New Kiruna: dynamic urban conversion and economy; lively and safe urban environment; easily accessible and walkable urban environment; mixed and socially cohesive urban environment; strong identity and architecture; interplay between the city and nature; energy- and resource-smart environment. The plan, as the practical sphere of the transformation, reflects non-local designers' ideas and a lack of sufficient communication with local planners. For example, dense structures might not be the right answer for Kiruna, as creating a grid city would not work in the arctic climate. The collaboration between planners from Kiruna and Stockholm contained conflicting ideas, because they were seemingly used to working in different contexts:

*"That is also one thing we struggle with because we have many architects coming from the south part of Sweden and they do not understand how it should work here. They come here every second month, they want to adapt ideas that do not work here. Kiruna people want to do things in certain ways. We need to adapt to new ideas gradually. That is also one problem with this dense city concept, we can see it already now; for instance how to handle snow? People want to have a clean city but you cannot do it in a dense city. So that is a challenge because planners tend to [understand... they] make assumptions. It has taken some time to understand, but now they started to understand. It is taking some time but I think we will find a common ground. I always told them that this is important for Kiruna people, we cannot have these compact blocks in the city center, you have to have passages between blocks. I do not want to create a copy of some other city, this should be 'Kiruna'"* (interview with Informant 7)

Kiruna's bold master plan includes urban design components that encourage a more economically diverse city, incorporates the development of environmentally sustainable infrastructure, offers a thoughtful method of gradual implementation (read more in Appendix D), and attempts to provide opportunities for the Kiruna identity to be preserved. Community participation is seen as a crucial pillar of the transformation process and there is strong awareness of its importance in the plan, and among planners, for creating a democratic city (interview with Informants 1, 5 & 9). This recognition was highlighted by introducing a "Kiruna dialogue," with the engagement of an anthropologist, to talk to people and understand how their wishes can be spatialized. This was a reaction to the above-mentioned issue that residents were reluctant to raise their voices and share their views on the move.

The social aspects of the transformation, raised in the plan for a democratic process, are valuable, yet do not make any impact on the spatial outcome. Residents' engagement in the process is aimed at having an informative process that justifies decisions and defeats criticism, rather than a participatory process that involves diverse residents and their opinions. The truth is, the question of whether or not Kiruna's master plan is a good one has yet to be determined, because the plan as it is currently written will not be completely fulfilled until 2100. Kiruna's master plan should continue to be examined and evaluated throughout its stages of implementation, because the lessons learned from its implementation have global significance.

The newly designed city centre appears to be far from the old one: scattered buildings and shops in the old city centre have now been replaced by a speculative square, with the new City Hall in its corner where commercial streets, as supposedly social spaces, are conjoined to this centre. The design of the new city centre clearly addresses the limitation of 'relocation,' as it means rebuilding a 'new' city 'spatially and socially' for 'old' citizens, with a few cultural and historical elements of the old Kiruna. The new City Hall, illustrated in Figure 4, is the first building in the new city, as a respectful and sensitive gesture to Kiruna's old city hall, a symbolic building that during 55 years became known as "Kiruna's living room," and had to be demolished. The clock tower was moved, for symbolic reasons, to the new City Hall building.

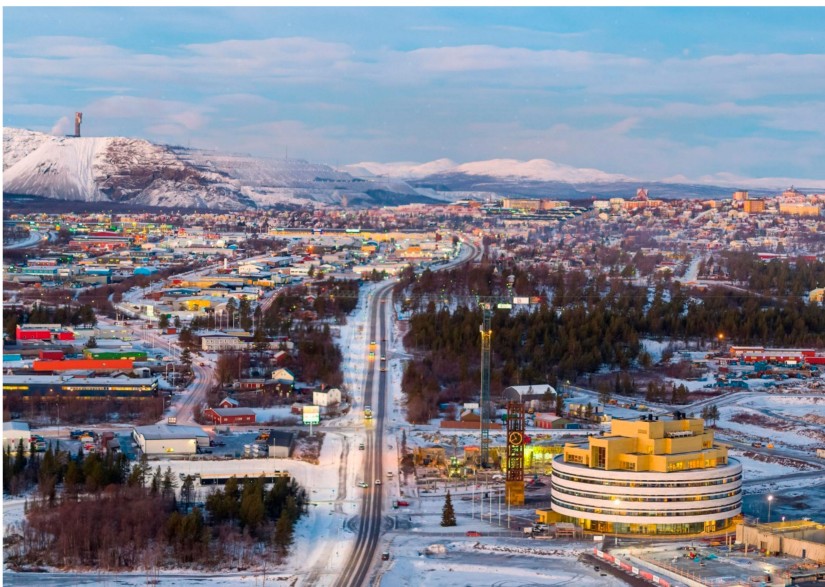

**Figure 4.** The newly built Kiruna town hall, with the old clock tower standing next to it. The old Kiruna and its mining landscape are in the background. Credit: Peter Rosén.

To enable the move within a short period of time, future designs and plans were used as a strategy to shift the focus from the uncertainties of the process to a positive outcome, and to legitimize the relocation process in a broader perspective. Design was used as a tool

to assure Kiruna's citizens of a better future to come, a 'politicized object' that could be used to avoid criticism and legitimate long-term decisions and short-term strategies [22]. The media had an important role in showing the municipality's capability in handling the process and in creating an overall positive expression about the relocation.

## 4. Discussion

Extractive industries and relocation are common trends in shaping urbanization in the north and elsewhere and, thus, Kiruna is not the first example. As long as natural resource extraction continues to be a driving factor of the economy, land use change and urban transformation, it is difficult to discuss human-induced disturbances on the natural and social environment in the context of sustainability.

Our attempt to integrate a three-sphere framework (personal, political and practical) for understanding the sustainability of Kiruna's urban transformation helped us investigate the project's process in these spheres and understand what shaped and dominated the narrative of relocation as well as its process. We have found partial responses in the framework with regard to the social and cultural values of the personal sphere that have not always been reached by the practical solutions and plans. By understanding the time plan of the mine and relocation, we found how the municipality's responses were guided towards more practical responses, thus reducing the sustainability of the project's outcome, which is discussed in the following subsection.

### 4.1. "Kiruna-4-Ever" for Whom?

As is often claimed, Kiruna's relocation is considered to be one of the biggest urban transformation projects in recent history [29]. The promotional video of the winning proposal, submitted by White Architects, in collaboration with Ghilardi + Hellsten Architects, and called "Kiruna-4-ever," starts with the statement that *"Kiruna was created as a town in 1900 by the state-owned mining company LKAB,"* thus setting a tone that emphasizes the town's mining identity, one established by LKAB [41], against a background that represents the fact that it was a land that Sami used for reindeer herding. The plan claims to be centered around people, who have voiced their opinions; on the basis of the information provided by them, they have in turn influenced the creation of the new city center's spatial form. This is a response to the Municipality's intention to make the plan a success story, using it as a tool to fulfill their claim of having developed the world's most democratic urban transformation, which would require making every single resident's view on the relocation heard. A concerned resident of Kiruna reflected upon the contrast:

> *"The main view about the move among the residents is that there has been this grand marketing of the whole city move that put the expectations really high. It was almost like it was a pity that for all the other cities in Sweden that they did not have to move as this was going to be such a wonderful thing and such a great opportunity. You could not help but think immediately—and I also think many people did that—it is typical Kiruna, in a way, marketing itself. We could not do it any other way than telling the world that this is the most fantastic thing"* (interview with Informant 11)

Narratives defined by the mine, within the political sphere, were overall accepted by Kiruna's residents, as their dependency on the mine—mentally and economically—is high. A Kiruna resident expressed his feelings around this dependency in a form of fear:

> *"We are used to live with a fear that the mine will close down one day. When the news came in 2004, there was absolutely no discussion by the Municipality, by the mining company, or by the media, whether we should move or not. There was almost no one who put that question. My feeling was that we were quite few people who would like to talk about whether we should move the city or not. If you would put this question on the table, you know what response you would get; they would say of course we move, the city is so dependent on the mine. It was also presented to us that the mining company was doing so well so it would go on for another 20–30 years and it is very good news for*

*Kiruna so that the future is secured for many years to come. Moving the city was not a big thing for the news, it was important to keep the mine running. It felt like those surveys were conducted just to show that they asked us, to show that it is democratic, to have us as alibis. But I think it is really, really hard to accomplish a true democratic process around it. You are always dependent on experts, the mining company of course and the other regulations. There is a very, very small window for an ordinary citizen to make an influence"* (telephone interview with Informant 11)

As the residents expressed in their responses, the rhetoric supporting the relocation presented the official optimism and was strongly dominant. Residents' collective memories and values, within the personal sphere, have been overlooked during the process and the plans failed to respond to their emotions:

*"I think there are a lot of problems about the whole 'prodigality' but generally they, I mean the Municipality, city planners and the mining company, really never want to talk about that... That is an issue and I think this whole process would have been more understanding and less rumours, less opposition, if they have been more . . . [pauses], everything that we lose, the surroundings, we should have talked more about these. This is a mourning process we have to go through, we are losing the city there, that many people loved and maybe hated at the same time. This is a big thing that affects people; you can see that there is a big nostalgia in Kiruna about how the city looked in history, I think that is something we should be recognizing. Especially the planners, but they are very busy putting up this picture of how good this is"* (telephone interview with Informant 11)

Even though residents do not necessarily see other options and might have also voted to move the city, some of the reactions reveal that possibly they would have preferred to have been asked:

*"We think that it should have been a big discussion, the answer would have been the same (the result would still have been to move) but it would have been good for Kiruna. We could discuss what would be the value in keeping this city, with this unique city planning, a city that is more than 100 years old with all the buildings. I think it is a big value that we are losing. Could we have something else? Would the tourist industry foster more just from keeping the city for example? I think all those things should have been discussed but it never came to the fore. I think that is a pity"* (telephone interview with Informant 11)

The plan creates a strong sustainability narrative that emphasizes the social dimension. However, Kiruna was founded on Sami land by the LKAB company around its prosperous mine ([42], p. 107) and, surprisingly, there is no reference made to the history of the indigenous Sami people who inhabited the land long before Kiruna was established. Kiruna's dominant mining profile led to exploitation of Sami's reindeer grazing lands and forced adaptation to industrial and urban expansion. As a result, the Sami population has gradually been marginalized ([42], p. 107).

If "Kiruna 4-ever" is to harness and foster diversity and give voice to different opinions and options, then the history, the images and voices of indigenous people could also be present in the promotion and branding of the strategy, as they are part of the town's history, present and future. Hence, past, present and future relations are not acknowledged or defined in a concrete way. A Sami from Kiruna describes the process from his perspective as the following:

*"It started with a meeting where invited stakeholders were present. Samis, Trafikverket (Swedish Transport Administration), the County Board, the Municipality, and other big companies were attending these meetings every second month. Everybody was allowed to discuss. There were big companies but we were also there and able to listen at least. It was more about the relocation and the railroad. It was a very open meeting and everybody listened to each other. Those meetings were really good but they ended in 2012*

*when the new railroad was opened. Then they quit these big meetings about the big city transformation. Since then there have never been meetings like this in which we could say how we as Sami community have been affected and what we want. Now when they are really moving the city, getting a new city hall, and everything is located in the new industrial area, so it will just keep continuing since those things need new areas, but at least it is better that they continue digging in the same area rather than opening a new one somewhere nearby"* (telephone interview with Informant 12)

Sami people do not hold aboriginal land rights in Sweden. During our site visit to Kiruna, a representative from a Sami village stated that Samis have the right to use the land for reindeer herding but they do not have the right to own the land:

*"When the mine expands further, we need to move away. If the mine expands even more, then we do not have any more place to go"* (overt observation during group visit to a Sami village, June 2015)

The Sami people we met expressed their concerns regarding their position in Swedish society and how their land use has been marginalized by the industrial activities for over a century. They stated that the understanding of indigenous rights is improving but certainly not at a level that they would wish it to be. LKAB is putting effort into adjusting their mining operations according to the reindeer herding season in order not to destroy the reindeers migration routes. However, the Sami have had to struggle to get their voices heard:

*"These mining activities have been affecting us, we are against it but we need to be careful how we express our ideas about it, otherwise the whole Kiruna residents might turn against us. It is basically LKAB says what is going to happen and then everybody has to accept it. And of course it is tricky when it is a state-owned company. We signed an agreement with LKAB in 2013. They are not allowed to destroy the land and make reindeer herding harder for us with their ongoing mining. They are not allowed to affect us more than they have already done. I am not allowed to talk much about this agreement, but if they need to do something affecting our reindeer herding, then they need to compensate us by maybe opening a new migration route for the reindeers. By that way, they compensate us. We were the ones pushing for this kind of compensation because they have been here for 100 years and we were never compensated or have never been listened to. Because it is such a big company, they have different parts, we have to sit and negotiate sometimes several days a week, with different parts of the company and nobody really... well, they have this bad picture of us and did not really respect us. We worked for years and negotiated before signing this agreement. It was really bad before and it is not that bad now. But this agreement is only with the LKAB, not with the municipality or other authority. If someone other than LKAB wants to build a road, it will again be a challenging situation, so in that sense our rights are not protected. It is still a struggle to maintain the land we have left"* (telephone interview with Informant 12)

The claims of the Sami residents and their lack of a collective identity in the new plan shows the contradiction in claiming the democratic and inclusive aspects of the process. The plan is a political tool for Kiruna Municipality and reflects a disconnection between personal and practical spheres.

Framing is the key issue at hand here. Our interviews with residents drew attention to the issue of inertia, in other words, status quo bias [43]. With frames being powerful due to the *"idea . . . that choices depend, in part, on the way in which options are stated"* and that in this way decision-makers become somewhat passive in relation to their selection ([44], p. 36). The choice about whether to move Kiruna seems to be framed to an extreme extent—there is only one option. The mining industry holds economic and political power within the town, and has clear, uncompromising interests in expanding mining activities, and this has been the reality that residents understand clearly. However, the framing of the narrative places people in the planning process; in fact, they were only involved in the design process, namely the product of the decision.

*4.2. Partial Responses to the Transformation Process*

The narrative created around the transformation claims that there is a connection between the three spheres; however, as the result of the previous sections indicates, we consider that these practices were partial responses to emerging problems. Uncertainty in the transformation process has been part of residents' everyday life, ever since the decision concerning the relocation was announced and accepted. The framed narrative around Kiruna's relocation is not a transformation of different spheres. It is mostly from the political sphere, with the hope of making change in other spheres, i.e., the practical and personal spheres, without enough interaction between them. Hence, Kiruna's urban transformation was presented in its political sphere as "unavoidable", to enable the relocation and delimit expressions of uncertainty.

Once the decision for the move was in place, the framing took a turn in the personal sphere, where individual and shared values are contained. This phase highlighted opportunities that this transformation would bring; a better planned, more sustainable and welcoming city for all. The response was to aim for the public's participation and a democratic planning process carried out by the decision-makers at the very center of the narrative. Here, there was value creation for the residents, where they were part of creating solutions collectively and could influence the type of solutions to be considered in the practical sphere. Previous research on the topic reflected upon the public discourse by engaging the concept of ideology to describe the 'images' produced in the media during 2006–2008, before the location of the new city was introduced [16].

Meanwhile, the actual relocation was discussed and, with regard to its site, there was a shift in responses from the personal to the practical sphere of the transformation. The spatial production and design of the new Kiruna, presented through strong images of its future, provided a rigid practical response. There was also an effort by the designers to connect practical and personal spheres. Included in the plan is a promise that emphasis is being placed on the importance of dialogue with the people living and working in Kiruna, and on providing tools for people to take part in the development process and contribute to the changing of their city. Yet that connection was a partial response to residents' spatial needs, while their emotions and uncertainties in the transformation process were not systematically considered (lack of engagement of the political sphere). Sandberg and Rönnblom [22] draw attention to the importance of emotional positions in the public discourse during 2012–2014.

*4.3. A Unique Transformation?*

Kiruna's urban transformation is addressed as a unique phenomenon, in terms of its scale, as nowhere in the world has such a large community had to be relocated because of an industrial operation ([7], pp. 93–95). Planners who were tasked with planning the new Kiruna had no examples upon which to base Kiruna's transformation planning. In Germany, 120 villages have been relocated since the 1920s, due to coal mining, but only a few hundred inhabitants per village were affected ([7], p. 95).

Parts of Kiruna's transformation could be replicated in terms of its new city center's design, combining both new and old physical and cultural elements. However, its financial model is unique, because the mining company is responsible and forced by the Swedish Mining and Mineral Act to cover the costs related to relocation. Besides their financial contribution, they are leading the implementation process, meaning direct engagement with the municipality and the residents, on a daily basis. The new highly profitable iron ore vein under the city center is also seen as one of the contributing factors to the uniqueness of this case.

The financial situation is usually different for the other relocation cases around the world, even within the same region, as in the case of Gällivare, a town 120 km south of Kiruna, and part of the county of Norrbotten, with more than 8000 inhabitants. The compensation policies for the relocation of inhabitants of Gällivare Municipality do not offer the same conditions as in Kiruna, and in some respects are missing. Malmberget, also

within the same county of Norrbotten, has faced the same destiny as Kiruna, and even though it, too, extracts iron ore for LKAB, it experienced a different process. The mining damage forced the town center's relocation during the 1960s; it was rebuilt with new buildings, except for the old church, which was moved to another location. Malmberget is referred to as a negative contrast, to illustrate the need to transform the city before the physical effects of mining manifest in the city and cause a rushed relocation [45]. Today, residents of Malmberget are not satisfied with how the process of moving is carried out; they claim that they are not compensated in the same way as the residents of Kiruna. Gällivare also does not attract the same attention as Kiruna. This justifies Kiruna's political approach in branding the relocation as a relatively unique case and compared to other local and global examples, which have not included compensation of affected inhabitants as part of their political agenda [46–48]. Extractivisms and their temporary value creation have been criticized globally, particularly in relation to capital accumulation [49]. Kiruna's transformation is an exceptional case in the way that its plans claim that they will diversify the future economy and reduce future dependency on mining activities. The plan is concerned with such independence, but also because the extension of exploitation activities is also uncertain (see [50] for updates from the mining company about the uncertainties of the mining process). This paper also demonstrates the indications for the changing nature of planning and the roles of planners, particularly in transformation of the personal and collective meanings in the urbanization process. Planning becomes more development-oriented and thus changes the role of planners to being collaborators or facilitators. This shift makes it difficult for spatial planners in Sweden, in particular, to work towards sustainable urban futures [51].

Kiruna's relocation is a unique and extreme case study, as the city and its residents are experiencing massive physical and social changes in a relatively short time. The case study of Kiruna is also context-specific, being tied as it is to an arctic climate, and thus the results cannot be generalized globally, even in Sweden within the same region of Norrbotten, as we discussed above in the situation of Malmberget. Our study sheds critical light on the branding of Kiruna's transformation as the most democratic one in the world, while highlighting partial political responses in addressing residents' values and needs. This is despite the fact that Kiruna is a relatively unique case, globally, because of its financial compensations and provisions of alternatives for the future homes of its residents [49].

## 5. Conclusions

This paper focuses on the relocation of Kiruna, a city in the arctic region of Sweden, whose inhabitants are affected by the mining activities of the largest iron ore producer in Europe. It raises questions about the notion of the sustainability of Kiruna's urban transformation, particularly by investigating its narrative and the driving forces behind the relocation. The media coverage both within and outside Sweden framed this relocation as a best practice and the most democratic urban transformation in the world. This research is an effort to provide a critical, multi-voiced narrative of transformation and understanding its sustainability through the interaction among the key actors involved within such a complex and uncertain process. The findings show the partial responses between these actors and the spheres of transformation, although they used rhetorical language to communicate the process, stressing the importance, uniqueness and magnitude of this transformation [33]. We highlight that these findings are limited to the period of the study (2015–2018), while the relocation is an ongoing project with constantly reshaping narratives. The initial narrative was constructed to divert attention from possible discussions of the process, by focusing on the uniqueness of the outcome. Packaging the whole urban transformation as a participatory planning process and a good opportunity to create a more sustainable city creates a paradox. Sustainability was used in the planning documents for the future city at a technical and practical level, rather than as a lens to analyze the transformation process.

In the foregoing, we discuss the limitation of spatial planning as a public sector instrument, for not having much other choice than to be flexible and pragmatic in order to

sustain Kiruna's economy and its residents. However, Kiruna's spatial (re)production is a long-term process and calls for a multi-voice understanding of its transformation process and better and stronger interaction between the residents, the local authority, planners and designers. We also discuss the impression that Kiruna's urban transformation cannot fit easily within the critiques that analyse this project either through the lens of market-led spatial production, or as a best practice for its democratic processes. The former critique is not fully valid in the Swedish welfare system, as the state-owned mining company is by law responsible to pay for the relocation and compensate affected residents. This also highlights how the citizens gained trust in the government's decisions and the fact that the mining company will take care of their future. Although Kiruna's transformation has not been fully democratic, the latter critique is not valid without framing this practice in a broader perspective. We discuss the claim that Kiruna's urban transformation is a unique and context-dependent project rather than a best practice, as it is tied to a special geography, that of the arctic, where the natural environment tends to be a key part in the considerations informing the decision-making process.

Our theoretical framework applies sustainability as a lens to assess the transformation process and suggests that achieving sustainability requires the connection between political, practical and personal spheres. It recommends paying further attention to the political sphere and the constraining of planning engagement in development-led transformation processes, and in connecting the values between personal and practical levels. Future research is needed to investigate how transforming systems and structures in the central political sphere could help to achieve a sustainable transformation.

**Author Contributions:** Conceptualization, A.T.D. and E.K.; methodology, A.T.D. and E.K.; validation, A.T.D. and E.K.; formal analysis, A.T.D. and E.K.; investigation, A.T.D. and E.K.; data curation, A.T.D. and E.K.; writing—original draft preparation, A.T.D. and E.K.; writing—review and editing, A.T.D. and E.K.; project administration, A.T.D.; funding acquisition, A.T.D. All authors have read and agreed to the published version of the manuscript.

**Funding:** This study was supported by the Fulbright Arctic Initiative Program 2015–2016, and Nordregio.

**Institutional Review Board Statement:** Ethical review and approval were waived for this study, due to the internal review made at the department. The design and content of this study do not require an application to the Swedish Ethical Review Authority according to the national legislation. The study neither includes medical treatment, bio samples, activities aimed for physical or psychological influence, including risk for being harmful on the status of the research person, nor sensitive personal or crime data.

**Informed Consent Statement:** Informed consent was obtained from all subjects involved in the study.

**Data Availability Statement:** The data presented in this study are available on request from the corresponding author. The data are not publicly available due to privacy reasons.

**Acknowledgments:** We would like to thank all the respondents in this research for their considerable input and time. We thank three anonymous reviewers for helpful suggestions to improve the paper. We also thank Professor Hans Westlund for his comments on the earlier drafts of this paper and Tigran Haas for his feedback.

**Conflicts of Interest:** The authors declare no conflict of interest.

### Appendix A

Between 2004, when LKAB submitted their proposal for changes, and 2007, when the municipality's new plan was launched, the company's impatience led it to develop an alternative plan for the city center's relocation. LKAB hired an architect and infrastructure consultants who presented a spatial planning proposal called 'New Kiruna' that included winding streets along mountain slopes, tropical garden and an indoor skiing hill under a glass roof ([36], p.43). While "New Kiruna' offered a bold vision for the town, the plan

created confusion in terms of responsibilities for urban planning and led to opposition by the municipality: the local planning administration had suggested that the new parts of the town be located to the east of the town, whereas the company's proposal located these to the northwest ([36], p.47). The municipality ultimately decided, as opposed to LKAB's proposal, that the city center would be located to the east of the town. Kiruna Municipality indeed demonstrated its integrity, in terms of being a decision-making authority for planning of the land, although, in contrast, the mining company's very initiative in proposing a development plan demonstrated its power to steer planning for the future development of Kiruna.

## Appendix B

**Table A1.** List of respondents' roles and details of conducted interviews.

| | Informants' Details | Interview Type and Date |
|---|---|---|
| 1 | Planner, Planning and Development Manager, Kiruna Municipality | Questionnaire responded to in Swedish, sent via e-mail, 1 January 2018 |
| 2 | Planner, Planning Architect, Kiruna Municipality | Questionnaire responded to in Swedish, sent via e-mail, 1 November 2018 |
| 3 | Planner, Kiruna Municipality | Face-to-face interview, 30 August 2016 |
| 4 | Politician, Kiruna Municipality | Face-to-face interview, conducted in Swedish, 29 August 2016 |
| 5 | Planner, Social transformation, LKAB | Questionnaire responded to in Swedish, sent via e-mail, 1 November 2018 |
| 6 | Planner, Urban transformation, LKAB | Questionnaire responded to in Swedish, sent via e-mail, 3 November 2018 |
| 7 | Planner, Architect and Project Manager, Urban transformation, LKAB | Face-to-face interview, 29 August 2016 |
| 8 | Planner, Sustainability Division, LKAB | Telephone interview, 17 April 2017 |
| 9 | Architect, White Architects | Face-to-face interview, 23 November 2018 |
| 10 | Academic working within Kiruna Sustainability Center, Luleå University of Technology | Telephone interview, 16 September 2016 |
| 11 | Resident, teacher, living in Kiruna Municipality | Telephone interview, 23 January 2017 |
| 12 | Resident, Sami from Kiruna | Telephone interview, 28 June 2017 |
| 13 | Resident, Sami from Kiruna | Face-to-face interview, 29 August 2016 |
| 14 | Resident, Shop-owner in Kiruna | Face-to-face interview, 30 August 2016 |
| 15 | Resident, Shop-owner in Kiruna | Face-to-face interview, 30 August 2016 |
| 16 | Resident, Taxi driver working in Kiruna | Face-to-face interview, 28 August 2016 |

## Appendix C

Compensation has been handled according to the Mining Act, which stipulated that the affected residents would be provided with two options: they could be offered either a comparable house with an equal value to their present one, or a sum of money that equals its market value, plus 25 percent [38].

## Appendix D

In their proposal, "Kiruna 4-ever," White Architects introduced three tools for the transformation process: the 'Kiruna dialogue'—a continuous citizen dialogue that informs the process, adds quality to the design, and gives the relocation a democratic platform; the 'Kiruna Biennale,' which invites people to bring experiences from other cities to Kiruna and gives the city a chance to learn from others; and the 'Kiruna Portal,' a virtual and physical

meeting place, offering a storage space where the old city can be re-used and transformed into the new [6,52].

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
