# Peer review of "Reframing Kiruna’s Relocation—Spatial Production or a Sustainable Transformation?"

_sustainability, doi:10.3390/su13073811_

Round 1
Reviewer 1 Report
Like other research strategies, the case study is a way of investigating an empirical topic by following a set of pre-specified procedures. The case of LKAB is either the critique of market-led spatial production or best practice for its participatory processes. Authors argue that a multi-voice narrative approach is necessary for the sustainability of Kiruna's transformation. This study, as a single case study, is justifiable under certain conditions. First, if it represents the critical case in testing a well-formulated theory, the single case can be used to determine whether a theory's propositions are correct or whether some alternative set of explanations might be more relevant. Second, whether the case represents an extreme or unique case, this has commonly been the situation in a single case worth documenting and analyzing. Third, whether it is a revelatory case, this situation exists when an investigator has an opportunity to observe and analyze a phenomenon previously inaccessible to scientific investigation. This case study examined the structure of an iron industry or the economy of a Kiruna city. It illustrated an investigation to retain the holistic and meaningful characteristics of real-life issues by interviewing stakeholders and residents in Kiruna. Nevertheless, this study needs to include the research methodology process of case studies. Although the topic and voice-narrative approach are somewhat fresh and meaningful, it looks like a reporter's journal article.Author Response
The response can be found as an attachment.

Reviewer 2 Report
The manuscript “Reframing Kiruna’s relocation – spatial production or a sustainable transformation?" presents a very interesting issue of city relocation, which constitutes a certain contribution to a sustainable spatial policy. In my opinion, this paper suits well into the scope of the Journal. However, I noticed some methodological weaknesses.
My comments mainly concern the structure of the article, some inconsistencies and inaccuracies in the presentation of the research and methodological issues. I hope the consideration of some of them will make the manuscript better readable and more transparent when it comes to interesting topic that presents.
However, I have some suggestions that may improve the article.
SPECIFIC COMMENTS:
I have noticed some editing errors. This editing part requires a certain improvement.
There is a lot of direct quotation from the interviewees. For this reason, the reader, sometimes, may lose the sense that he is reading a scientific article.
Keywords:
- Please find such words which are not in the title, this way search engines of the web will find your manuscript with higher probability. There are also keywords – like Swedish planning - that are not related to the essence of the article.
Introduction:
- The article lacks a clearly defined and unambiguous formulation of the aim of the work. Perhaps adding research questions would clarify the indication of the purpose of the research.
- The introduction does not strongly indicate innovation in the conducted research. I encourage the authors to highlight there the innovative solutions developed in this study.
- The separation of the Swedish Planning process chapter is, in my opinion, unjustified. It also disturb the continuity of the introduction about Kiruna. I suggest incorporating this information into the content related to the spatial planning process in Kiruna, as a form of outlining the principles of spatial planning in Sweden in relation to the analysed area.
- Social consultations are an important stage of the described process. In the article I miss an indication of how public consultations are carried out in Sweden. How long did this process take in Kiruna's case?
- Figure 1 is not of good quality
Materials and Methods:
- An interview with representatives of various social groups was the basic material for the study. In my opinion, the article lacks:
- precise indication of what questions were specifically asked
- precise indication of all respondents - from what groups? how many people?
- precise indication of the number of inhabitants with whom the interview was conducted
- on what basis the respondents were selected
- Due to the above, my doubts are also raised by the fact, if the survey was not carried out on a too small sample of residents. In this case, I see a some methodological weakness in the study.
Results
- Inconsistency in providing sources of information (for interviews) - sometimes the form of contact is given, the date, and sometimes not.
- Figure 3 - should appear in the text after it is recalled, not before.
Discussion:
- The discussion is too short compared to the extensive results chapter.
- In Discussion section, the authors should discuss the results of the research and compare them with other studies. In the discussion should be cited appropriate literature. Try to discuss the results with relation to other cities. I suggest rewriting the discussion and focusing on comparing the results with other cases.
- Authors write - in abstract - that Kiruna's transformation is considered sustainable throughout the world. The text lacks the elaboration of this thread - an indication of the sources of such opinions. It is worth discussing such issues in the discussion chapter - to compare it with your own results
Conclusion:
- There is no conclusion section. Manuscript shouldn't end up in discussion. In my opinion, some of the discussions chapter are actually conclusions.
Author Response
The response can be found as an attachment.

Reviewer 3 Report
This is a very interesting paper that addresses the process of urban transformation of an artic city in Sweden. The authors do a fantastic job in applying a three-sphere conceptualization to the the relationship between sustainability and Kiruna's path towards a major iron ore producer in the EU.
I suggest the authors make some minor revisions to the manuscript. First, the study draws upon the interviews with 13 stakeholders. It would be the best if the authors can add a table to explicitly mention how many of them are planners and how many are residents. Second, a map that can show the location of Kiruna in the region would be needed. Third, would the theory of resource curse be applicable in this case?
Author Response

(The authors gave the same response as above.)

Round 2
Reviewer 1 Report
Minor check is required, such as typos.
Author Response
Dear Reviewer 1,
Thank you very much for your comment. Our response is attached.

Reviewer 2 Report
Dear Authors,
Thank you for your answers to my comments - they are very comprehensive. I appreciate the effort put into improving the quality of the article. Still, I have a few minor comments:
1) I still suggest improving 'Keywords'. In my opinion, the words 'Kiruna', 'relocation' should not be included in the keywords because they appear in the title of the manuscript. In addition, 'urban transformation' appears twice in keywords. I also have doubts about the use of 'best practice'. Moreover, I suggest that you should leave only one of these two phrases: 'urban planning' or 'planning practice' because they are very similar in terms of meaning.
2) In my opinion, referring to literature in sentences concerning the formulation of the aim of the work is redundant and unnecessary.
3) In the received new manuscript all the figures are of poor quality - they are illegible.
4) Table 1 presents the relevant information about the interviews conducted. However, I suggest placing it in the Appendix section
Author Response
Dear Editor,
Thank you very much for your valuable time and comments. Our response can be found attached.
